



# Analog Data Assimilation for the Selection of Suitable General Circulation Models

Juan Ruiz[1], Pierre Ailliot[2], Thi Tuyet Trang Chau[3], Pierre Le Bras[4,5], Valérie Monbet[6], Florian Sévellec[4,7], and Pierre Tandeo[5]

[1]Centro de Investigaciones del Mar y la Atmósfera, Facultad de Ciencias Exactas y Naturales, Universidad de Buenos Aires, CONICET-UBA, Buenos Aires, Argentina, UMI-IFAECI (CNRS-CONICET-UBA), Buenos Aires, Argentina
[2]Univ Brest, CNRS UMR 6205, Laboratoire de Mathematiques de Bretagne Atlantique, France
[3]LSCE, IPSL-CEA Saclay, 91191 Gif-sur-Yvette cedex, France
[4]Laboratoire d'Océanographie Physique et Spatiale, IUEM, Univ. Brest, CNRS, IRD, Ifremer, Brest, France
[5]IMT Atlantique, Lab-STICC, UMR CNRS 6285, F-29238, France
[6]INRIA & Univ Rennes, CNRS, IRMAR - UMR 6625, F-35000 Rennes, France
[7]Ocean and Earth Science, University of Southampton, Southampton, United Kingdom

**Correspondence:** Juan Ruiz (jruiz@cima.fcen.uba.ar)

**Abstract.** Data assimilation is a relevant framework to merge a dynamical model with noisy observations. When various models are in competition, the question is to find the model that best matches the observations. This matching can be measured by using the model evidence, defined by the likelihood of the observations given the model. This study explores the performance of model selection based on model evidence computed using data-driven data assimilation, where dynamical models are emulated using machine learning methods. In this work, the methodology is tested with the three-variable Lorenz' model and with an intermediate complexity atmospheric general circulation model (a.k.a. the SPEEDY model). Numerical experiments show that the data-driven implementation of the model selection algorithm performs as well as the one that uses the dynamical model. The technique is able of selecting the best model among a set of possible models and also to characterize the spatio-temporal variability of the model sensitivity. Moreover, the technique is sensitive to differences in the model dynamics which are not reflected in the moments of the climatological probability distribution of the state variables. This suggests the implementation of this technique using available long-term observations and model simulations.

## 1 Introduction

Data assimilation (DA) methods aim to provide the best estimation of the state of a dynamical system based on a set of noisy and partial observations (see Carrassi et al., 2018; Reich, 2019; Van Leeuwen et al., 2019, and references therein). Current state-of-the-art data assimilation systems are based on robust mathematical grounds, allowing to expand their use beyond their original aim. One application that has recently received increasing attention is the use of data assimilation methods for model optimization and model selection. The former is concerned with obtaining better estimates for model parameters and configuration with the ultimate goal of quantifying and reducing model error and dispersion in various applications (e.g., Schirber et al., 2013; Ruiz et al., 2013; Ruiz and Pulido, 2015; Lauvaux et al., 2019; Kotsuki et al., 2020). Model selection





aims at identifying the model which best describes a set of observations in a finite set of possible models. In meteorology and oceanography, model selection may be useful for example to find the best physics for forecast applications or for detecting the forcing terms that better explain the evolution of a dynamical system for attribution purposes.

So far model selection, in the context of data assimilation, seems to have received less attention than model optimization. In Hannart et al. (2016), the authors show that the fraction of attributable risk in the context of climate change (Pearl, 2000)

can be estimated as a by-product of two data assimilation systems. One of these systems is run with a model which includes forcing consistent with anthropogenic emissions, and another is run without considering those emissions. This approach was proven to be more sensitive to differences between the two scenarios at a lower computational cost than other available attribution techniques. One usual approach to conduct model selection is based on the computation of model evidence, which is defined as the log-likelihood of the available observations for a given model configuration (Reich and Cotter, 2015). Carson

et al. (2018) proposed to use model evidence to select a model consistent with data records in paleoclimate science. This work succeeds in both fitting conceptual models and identifying the one with the most appropriate orbital forcing to represent the glacial–interglacial cycle. Estimating model evidence, however, is generally complex for practical applications since geophysical models are usually high-dimensional and nonlinear. In such circumstances, it is crucial to develop and to implement data assimilation methods which allow accurate estimations of model evidence.

Carrassi et al. (2017) introduced the concept of Contextual Model Evidence (CME) which can be roughly defined as the log-likelihood of a set of observations over a short time period for a given dynamical model. Based on this approach, the model evaluation can be obtained by running several data assimilation systems (i.e. each one using a particular dynamical model) and then comparing their corresponding CME values. Metref et al. (2019) extended this idea to high dimensional systems and studied the impact of domain localization upon the computation of the CME in the context of an ensemble Kalman filter

(EnKF). A similar idea has been implemented by Otsuka and Miyoshi (2015) for the online optimization of a multi-model EnKF. They run a particle filter that assigns weights to each model configuration based on the likelihood of the observations given different model configurations. The approach successfully identifies the most accurate model improving the performance of both the assimilation and the forecast.

The articles cited above perform model selection using classical data assimilation methods where the different competing

models that represent the dynamics of a particular system must be solved several time at each time step. Recently, Tandeo et al. (2015); Lguensat et al. (2017) introduced the concept of Analog data assimilation (AnDA). This approach can be particularly beneficial in cases where the numerical model is not known explicitly or extremely computationally expensive (as it is usually the case with state-of-the-art numerical models of the climate system) or when such models are not available and only the observation dataset exists. In this case, AnDA can take advantage of existing long-term climate model simulations or observa-

tions and perform DA by emulating the dynamical model using nearest neighbor regression, also called analog forecasting in meteorology Lorenz (1969).

In this paper, we combine the AnDA method with the computation of the CME. The objective is to provide a proof of concept using numerical experiments on the low dimensional modified Lorenz-63 toy model (Lorenz, 1963) and the intermediate





complexity atmospheric general circulation model SPEEDY (Molteni, 2003). This proof of concept suggest the possible use of

existing model simulations to efficiently compare, based on observations, different models and physics.

The paper is organized as follows. Section 2 provides a brief review of model evidence and introduces an algorithm to compute CME using AnDA. Section 3 presents the numerical results and Section 4 presents final remarks and perspectives for future research.

## 2   Methodology

Model evidence measures the ability of a dynamical model $\mathcal{M}$ to describe a sequence of multivariate, noisy, and partial observations $\mathbf{y}_{0:K} = (\mathbf{y}_0, \ldots, \mathbf{y}_K)$ (from a sufficient long time in the past and up to time $K$). It is a useful tool to identify the model best fitting a set of observations in a list of competing models. In this section, after defining model evidence, we discuss its computation by combining data assimilation and analog forecasting.

### 2.1   Contextual model evidence

#### 2.1.1   Definition

Model selection or comparison is usually performed using the climatological Model Evidence (ME), see Metref et al. (2019) and references herein. It corresponds to $\ln p(\mathbf{y}_{0:K}|\mathcal{M})$, the global log-likelihood of the observations $\mathbf{y}_{0:K}$ for the dynamical model $\mathcal{M}$. This ME metric is roughly measuring the adequacy between the observations available up to time $K$ and the climatological distribution of the model. Though, this global metric is probably not descriptive enough for studying the model

performance over a particular time period or the transition between different states of the system.

Alternatively, Carrassi et al. (2017) proposed to compute the Contextual Model Evidence (CME) defined as the local log-likelihood on a short interval of time. More precisely, it is defined as

$$\mathrm{CME}_{k:k+h}(\mathcal{M}) = \ln p(\mathbf{y}_{k:k+h}|\mathbf{y}_{0:k-1};\mathcal{M})$$

where $p(\mathbf{y}_{k:k+h}|\mathbf{y}_{0:k-1};\mathcal{M})$ denotes the conditional (forecast) distribution of the observations $\mathbf{y}_{k:k+h}$ between times $k$ and $k+h$ given the observations up to time $k-1$ for model $\mathcal{M}$ and $h$ is the width of the evidencing window. As stated in Metref et al. (2019), a key difference between CME and ME is that the former takes into account the actual state of the system. This information is considered in the a-priori estimation of the state of the system at the beginning of the evidencing window which

is assumed to be approximately known. So, CME computes the evidence taking into account the context, which can provide a more detailed local evaluation of the system dynamics.

Remark that

$$\mathrm{CME}_{k:k+h}(\mathcal{M}) = \sum_{i=k}^{k+h} \ln p(\mathbf{y}_i|\mathbf{y}_{0:i-1};\mathcal{M}) = \sum_{i=k}^{k+h} \mathrm{CME}_i(\mathcal{M}) \qquad (1)$$

where $\mathrm{CME}_i(\mathcal{M}) = \ln p(\mathbf{y}_i|\mathbf{y}_{0:i-1};\mathcal{M})$ is the CME in the particular case where evidencing window is reduced to a single

time. This last quantity corresponds to the Logarithmic Score (also known as the Ignorance Score, see e.g. Siegert et al. (2019)





and references therein) of the forecast distribution $p(\mathbf{y}_i|\mathbf{y}_{0:i-1};\mathcal{M})$ associated to model $\mathcal{M}$. Also when $k$ is set to $0$ in the CME (i.e. there is no observation prior to the evidencing window), we obtain the ME.

CME cannot be evaluated directly because the observations $\mathbf{y}$ are often incomplete, intermittent and uncertain. To tackle these issues, a latent variable which represents the true state of the system is introduced leading to the following state-space model

$$\mathbf{x}_k = \mathcal{M}(\mathbf{x}_{k-1}) + \boldsymbol{\eta}_k, \tag{2}$$

$$\mathbf{y}_k = \mathcal{H}(\mathbf{x}_k) + \boldsymbol{\epsilon}_k, \tag{3}$$

where $\mathbf{x}_k$ denotes the latent state and $\boldsymbol{\eta}_k$ represents the model noise (i.e. the part of the system dynamics which is not represented by the numerical model) at time $k$. In Eq. (3), $\mathcal{H}$ is the observation operator, representing the link between the latent state $\mathbf{x}$ (i.e., what we want to estimate) and the observations $\mathbf{y}$. The additive term $\boldsymbol{\epsilon}_k$ represents observation errors.

For a state-space model $\mathrm{CME}_k(\mathcal{M})$ can be expressed as follows

$$\mathrm{CME}_k(\mathcal{M}) = \ln \int p(\mathbf{y}_k|\mathbf{x}_k)\, p(\mathbf{x}_k|\mathbf{y}_{0:k-1};\mathcal{M})\, d\mathbf{x}_k. \tag{4}$$

The forecast distribution $p(\mathbf{x}_k|\mathbf{y}_{0:k-1};\mathcal{M})$ which appears in the previous expression can again be decomposed into two terms

$$p(\mathbf{x}_k|\mathbf{y}_{0:k-1};\mathcal{M}) = \int p(\mathbf{x}_k|\mathbf{x}_{k-1};\mathcal{M})\, p(\mathbf{x}_{k-1}|\mathbf{y}_{0:k-1};\mathcal{M})\, d\mathbf{x}_{k-1}. \tag{5}$$

where the analysis distribution $p(\mathbf{x}_{k-1}|\mathbf{y}_{0:k-1};\mathcal{M})$ represents the estimation of the state of the system at time $k-1$ given all the previous available observations (i.e. the context which is usually provided by a DA system).

### 2.1.2 Contextual model evidence and the ensemble Kalman filter

In an ensemble Kalman filter (EnKF), the forecast distribution $p(\mathbf{x}_k|\mathbf{y}_{0:k-1};\mathcal{M})$ is approximated using a Monte Carlo approach performing multiple evaluations of the model $\mathcal{M}$, initialized from a sample of states drawn from the analysis distribution $p(\mathbf{x}_{k-1}|\mathbf{y}_{0:k-1};\mathcal{M})$. More precisely, a sample of the forecast distribution is generated as follows

$$\mathbf{x}_{(j),k}^f = \mathcal{M}(\mathbf{x}_{(j),k-1}^a) + \eta_{(j),k} \tag{6}$$

where $\left\{\mathbf{x}_{(j),k-1}^a\right\}_{j=1:N}$ is a sample of $N$ members from the analysis distribution at time $k-1$ and $\eta_{(j),k}$ is a realization of a stochastic process representing model imperfections typically drawn from a Gaussian distribution with zero mean and covariance $Q$ (see Tandeo et al., 2020). The different members $\left\{\mathbf{x}_{(j),k}^f\right\}_{j=1:N}$ of the forecast are used to approximate the first two moments of the forecast distribution through the sample mean

$$\overline{\mathbf{x}}_k^f = \frac{1}{N}\sum_{j=1}^N \mathbf{x}_{(j),k}^f, \tag{7}$$





and covariance

$$\boldsymbol{\Sigma}_k^f = \frac{1}{N-1} \sum_{j=1}^{N} (\mathbf{x}_{(j),k}^f - \overline{\mathbf{x}}_k^f)(\mathbf{x}_{(j),k}^f - \overline{\mathbf{x}}_k^f)^T. \tag{8}$$

When observations are available at time $k$, the state distribution can be updated based on the information provided by the observations. This update is performed based on the Bayes theorem assuming that both the observation likelihood and the forecast distributions are Gaussian and that their moments are well approximated by the sample moments. The update can be conducted in different ways, one possible approach is the so called stochastic EnKF update (Burgers et al., 1998) in which each member of the analysis sample is obtained as

$$\mathbf{x}_{(j),k}^a = \mathbf{x}_{(j),k}^f + \mathbf{K}_k \left( \mathbf{y}_k - \mathcal{H}(\mathbf{x}_{(j),k}^f) + \boldsymbol{\epsilon}_{(j),k} \right) \tag{9}$$

with $\mathbf{K}_k$ the Kalman filter gain defined as

$$\mathbf{K}_k = \boldsymbol{\Sigma}_k^f \mathbf{H}^T (\mathbf{H}\boldsymbol{\Sigma}_k^f \mathbf{H}^T + \mathbf{R})^{-1}. \tag{10}$$

where $\mathbf{H}$ denotes the tangent linear of the observation operator $\mathcal{H}$.

Eqs. (6-10) can be used sequentially to produce an estimate of the state conditioned on previous observations ($\mathbf{y}_{0:k-1}$) at each time $k$. Given an EnKF sequential cycle, based again on Gaussian assumptions, $\text{CME}_k(\mathcal{M})$ can be approximated as (Carrassi et al., 2017):

$$\text{CME}_k(\mathcal{M}) \approx -\frac{1}{2} (\mathbf{y}_k - \mathcal{H}(\overline{\mathbf{x}}_k^f))(\mathbf{H}\boldsymbol{\Sigma}_k^f \mathbf{H}^T + \mathbf{R})(\mathbf{y}_k - \mathcal{H}(\overline{\mathbf{x}}_k^f))^T - \frac{1}{2} \ln |\mathbf{H}\boldsymbol{\Sigma}_k^f \mathbf{H}^T + \mathbf{R}| - \frac{n}{2} \ln(2\pi), \tag{11}$$

where $n$ is the number of available observations at time $k$. Combining Eqs. (1) and (11) (also with Eqs. (6-10)) we can compute $\text{CME}_{k:k+h}$ from an ensemble Kalman filter based data assimilation system for an arbitrary time window $[k, k+h]$. When both forecast mean and covariance are well estimated, Carrassi et al. (2017) showed that this CME approach is able to detect the model candidates which best describe a given set of observations.

Ensemble Kalman filters usually suffer from systematic underestimation of the forecast error variance. To compensate for this issue an ad-hoc multiplicative inflation coefficient $\gamma$ is usually applied such that

$$\boldsymbol{\Sigma}_k^f \longleftarrow \gamma \boldsymbol{\Sigma}_k^f. \tag{12}$$

The inflation coefficient $\gamma$ can be estimated online using different techniques which are based on the work of Desroziers et al. (2005). In particular Li et al. (2009) propose the following estimation approach, in which at each time $k$ the inflation value is updated according to

$$\gamma_{k+1} = \rho \left( \frac{Tr(\boldsymbol{d}_{o-f}^T \boldsymbol{d}_{o-f} - \boldsymbol{d}_{o-a}^T \boldsymbol{d}_{o-f})}{Tr(\mathbf{H}\boldsymbol{\Sigma}_k^f \mathbf{H}^T)} \right) + (1-\rho)\gamma_k \tag{13}$$

with $0 \leq \rho \leq 1$ a time smoothing parameter, $d_{o-f}$ the difference between the forecast and the observations in the observation space and $d_{o-a}$ the difference between the analysis and the observations in the observation space.





## 2.2 Data-driven contextual model evidence

### 2.2.1 The analog ensemble Kalman filter

In a data assimilation system, the numerical model is required to propagate the information in time. Particularly, in the case of ensemble based assimilation techniques, the numerical model has to be run several times (once for each of the ensemble members and each time step). An alternative to this computationally intensive approach is to use analog forecasting. This method, initially introduced by Lorenz (1969) in meteorology, is a data-driven procedure which uses a catalog of historical data, most of the time corresponding to simulation runs or analysis (i.e., outputs of data assimilation procedures). The idea is to search in the catalog, using an appropriate distance, the nearest analogs of the initial condition from which we want to get a forecast. Then, the successors of these closest analogs are combined to get a probabilistic forecast. Analog forecasting is popular because of its simplicity and robustness. Several studies confirm these advantages in the context of environmental sciences (Barnett and Preisendorfer, 1978; Bannayan and Hoogenboom, 2008; Yiou, 2014; Atencia and Zawadzki, 2015; Ayet and Tandeo, 2018; Sévellec and Drijfhout, 2018).

The combination of data assimilation and analog forecasting has been proposed in (Tandeo et al., 2015; Lguensat et al., 2017) leading to the the Analog Data Assimilation (hereinafter AnDA) method. It is a flexible framework that can be adapted to a large set of problems. An interesting feature of AnDA is that it can be applied locally, without the need to approximate the full model. Also, as most DA systems, AnDA can handle complex observations with irregular spatio-temporal distributions as long as an appropriate observation operator is available. More precisely, AnDA corresponds to running a DA algorithm using the state-space model (2-3) with (2) replaced by

$$\mathbf{x}_k = \widehat{\mathcal{M}}(\mathbf{x}_{k-1}) + \boldsymbol{\eta}_k, \tag{14}$$

where $\widehat{\mathcal{M}}$ denoting the analog-based approximation of the the dynamical model $\mathcal{M}$.

An efficient statistical forecast operator used in Lguensat et al. (2017) and detailed in Platzer et al. (2021) is the local linear regression originally introduced in Cleveland and Devlin (1988). It consists in first searching in the catalog the M-nearest neighbors (i.e. the analogs) of a given state $\mathbf{x}$ along with their corresponding successors in time. Then, a multiple linear regression is fitted between the $M$ analogs and their successors. The coefficients of this regression are denoted $\beta(\mathbf{x})$ and $\alpha(\mathbf{x})$, where we stress that these coefficients depend on the state of the system. Note that this regression is nonlinear since the linear approximation is applied locally in state space and corresponds to a first-order expansion of the dynamical model (see Platzer et al., 2021).

The combination of analog forecasting, based on local linear regressions, and the EnKF leads to the AnEnKF algorithm where Eq. (6) is replaced by

$$\mathbf{x}_k^f = \underbrace{\alpha(\mathbf{x}_{k-1}^a)\mathbf{x}_{k-1}^a + \beta(\mathbf{x}_{k-1}^a)}_{\widehat{\mathcal{M}}(\mathbf{x}_{k-1}^a)} + \eta_k. \tag{15}$$





The model error $\eta_k$ is drawn from a mutlivatiate Gaussian distribution $\mathcal{N}(0, \tilde{\boldsymbol{\Sigma}}(\mathbf{x}_{k-1}^a))$ where $\tilde{\boldsymbol{\Sigma}}(\mathbf{x}_{k-1}^a)$ denotes the sample

covariance of the residuals of the multiple linear regression between the analogs and the successors (see Lguensat et al., 2017,

their Section 3a for further details). Once the forecast is performed using the analog technique, the analysis update can be

done as in the EnKF using Eq. (9). Multiplicative inflation (Eq. 12) could also be applied and estimated online using Eq. (13).

AnEnKF has been tested on toy dynamical models in Lguensat et al. (2017). Numerical results show that the performance of

the classic EnKF (i.e., using the true model $\mathcal{M}$) and AnEnKF (i.e., using analog forecasts $\widehat{\mathcal{M}}$), is almost the same when the

size of the catalog is large enough.

However applying efficiently analog forecasting for chaotic dynamics with strong nonlinearities and high dimensional spaces

is not always straightforward. As mentioned in Zhen et al. (2020), the analog space must be large enough to capture the dy-

namics of the system, but small enough to avoid the curse of dimensionality. This can be achieved using time-delay embedding

and projection in appropriate subspace. In this work, this issue is avoided by running AnDA locally on a reduced set of the

state-space variables (see Section 3.2).

### 2.2.2  Contextual model evidence and the analog ensemble Kalman filter

It is straightforward to combine the CME and AnEnKF procedures in the same data assimilation scheme. The procedure is

summarized in Fig. 1 and detailed in Alg. 1.

---

**Algorithm 1 Contextual model evidence using ensemble Kalman Filter and local linear regression forecasting**

---

  – Initialization: sample the first ensemble, $\left\{\mathbf{x}_{(j),0}^a\right\}_{j=1:N} \sim p(\mathbf{x}_0)$.

  – For $k = 1 : K$,

   **+ Forecast step**:

     - for each member $j = 1, \ldots, N$ propagate the previous analysis member using the analog forecasting operator Eq. (15).

     - compute the empirical mean forecast $\overline{\mathbf{x}}_k^f$ (7) and covariance forecast $\boldsymbol{\Sigma}_k^f$ (8) from the ensemble members.

     - apply the multiplicative inflation factor Eq. (12)

     - compute the contextual model evidence Eq. (11).

   **+ Analysis step**:

     - update the state distribution Eq. (9-10).

     - update the inflation factor Eq. (13).

---

In the next section, several models $\left\{\mathcal{M}^{(i)}\right\}_{i=1:L}$ are in competition and CME is used to identify those which best describe

the dynamics of the observations. Note that the time series $\mathrm{CME}_k(\mathcal{M}^{(i)})$ are computed independently for each model. The

combination of CME and AnEnKF have several advantages in this context. Firstly, it is a fast procedure, because it avoids to

run the different models at each time step and for each ensemble member. Secondly, model errors associated with the local

linear regression are considered and contribute to the forecast ensemble spread. Thirdly, it allows to compute the CME in





specific regions of the state space (e.g., in specific areas or integrated variables), only where observations are available. This last point will be discussed in Section 3.2 using the SPEEDY model.

## 3   Results

This section presents numerical results which show that the proposed methodology is able to identify the model which is the most compatible with a given set of observations. A first set of experiment is performed using a modified version of the

Lorenz-63 model. Then we focus on the more challenging intermediate-complexity atmospheric general circulation model SPEEDY.

### 3.1   Modified Lorenz-63 system

In this section, the method is tested on a modified Lorenz-63 model $\mathcal{M}^{(\lambda)}$ originally introduced in Palmer (1999). It is defined by the following system of differential equations

$$
\begin{aligned}
\frac{\mathrm{d}x_1}{\mathrm{d}t} &= 10\left(x_2 - x_1\right) + \lambda\cos\left(\tfrac{7}{9}\pi\right), \\
\frac{\mathrm{d}x_2}{\mathrm{d}t} &= x_1\left(28 - x_3\right) - x_2 + \lambda\sin\left(\tfrac{7}{9}\pi\right), \\
\frac{\mathrm{d}x_3}{\mathrm{d}t} &= x_1 x_2 - \tfrac{8}{3}x_3,
\end{aligned}
\tag{16}
$$


where $\lambda$ controls the magnitude of the external forcing term which is added on the first two components. The particular case $\lambda = 0$ corresponds to the classical Lorenz-63 model (Lorenz, 1963). Figure 2 shows two trajectories simulated from Eq. (16) with $\lambda = 0$ and $\lambda = 8$. When $\lambda > 0$, the additional forcing term "pushes" the trajectories towards the right wing and the left wing is less often visited than the right one, the opposite holding true when $\lambda < 0$. This behaviour can be clearly seen on the

right plots in Fig. 2 which show the number of times the system visits different regions of the state space.

A set of observations is generated using the classical Lorenz-63 model $\mathcal{M}^{(0)}$, referred to as the correct model hereafter. We conduct several experiments to check if the proposed methodology is able to identify that the observations were actually generated using the correct model $\mathcal{M}^{(0)}$ and not from another model in the list of competing models which corresponds to the modified Lorenz models $\mathcal{M}^{(\lambda)}$ with $\lambda = -8,\ldots,8$. In practice, the Lorenz model is integrated using a Runge-Kutta 4-5

scheme and the three components are observed with a time step $\mathrm{d}t = 0.1$. The observation operator is $\mathcal{H} = \mathbf{I}_3$ (the $3\times3$ identity matrix) and an additive Gaussian error with covariance $\mathbf{R} = 2\mathbf{I}_3$ is used. Hereafter each experiment is based on $K = 10^4$ data assimilation cycles. Unless stated otherwise, the AnEnKF is run using catalogs of size $T = 10^4$ and the evidencing window size is $h = 1$.

Comparing the left panels of Figure 3 shows that the EnKF is very sensitive to the ensemble member size $N$, especially

when the strength of the forcing in the modified Lorenz-63 model (i.e. the value $|\lambda|$ ) increases. Applying a multiplicative inflation factor (see last row of Figure 3) permits to solve this issue. Hereafter we discuss the results obtained when both EnKF and AnEnKF algorithms are run with $N = 20$ ensemble members and a multiplicative inflation factor is applied. This allows to obtain robust results for the EnKF at a reasonable computational cost. Remark that the AnEnKF is less sensitive to the ensemble size and that adding inflation do not seem necessary here (see right panel of Figure 3). This may be explained by the





adaptive data-driven procedure which is used to estimate the forecast error in the AnEnKF procedure. Figure 3 shows the mean CME for different values of $\lambda$. As expected, the CME has a larger mean value when the correct model $\mathcal{M}^{(0)}$ is used in the data assimilation procedures and decreases when the value of $|\lambda|$ increases. Both EnKF and AnEnKF give comparable results when multiplicative inflation (Eq. 13) is applied (see bottom panels of Figure 3).

    As an example, Fig. 4 (top panel) shows the analysis for the second component $x_2$ obtained when running the AnEnKF with

a catalog of the correct model $\mathcal{M}(0)$. It suggests that the AnEnKF is generally able to reconstruct the true state of the system. This can be assessed more precisely by computing the RMSE between the true state $x$ and the analysed state $x^a$ (0.50) and the mean coverage probability of the $95\%$ prediction interval ($84.26\%$). Then the AnEnKF was run with a catalog of the incorrect model $\mathcal{M}^{(8)}$. As expected, the RMSE is larger (0.72) and the mean coverage probability is also degraded ($63.86\%$), and thus less accurate estimates of the mean and the variance of the true state are obtained. Remark that these two quantities cannot be

computed in practical applications since the true state is not known. The bottom panel of Figure 4 shows that the CME obtained with the AnEnKF run with a catalog of the incorrect model are generally smaller than the ones obtained with a catalog of the correct model. It illustrates again that the CME computed with AnEnKF can be used to identify the correct model using only a sequence of noisy observations and catalogs of competing models.

    This is confirmed by the results given in Table 1. The CME associated to the correct model is larger than the one associated

to the incorrect model for $68\%$ of the assimilation cycles. Remark that this percentage of correct identification increases with the catalog size $T$ and seems to converge to the percentage of correct identification obtained with the EnKF ($68\%$) when $T$ becomes large. This is not surprising since using larger catalog provides a better approximation of the dynamical model and thus similar results when using EnKF and AnEnKF (see the discussion in Lguensat et al., 2017). Table 1 also shows that CME is more precise in identifying the correct model compared to the root mean squared error of the forecast in the observation

space defined as $\text{RMSE}_k^f = \|\mathbf{y}_k - \mathcal{H}(\overline{\mathbf{x}}_k^f)\|$. Remark that, according to Eq. (13), CME depends not only on the forecast error but also on the variance of the forecast error and this may help to identify the correct model. Also, to highlight the advantage of using data assimilation to identify the correct model we perform an experiment using only the pure analog forecasting (AnF), without data assimilation. For each time $k$, we select analogs based only on the available noisy observations and we propagate the state using the local linear regression approach. The RMSE of each forecast at time $k+1$, denoted $\text{RMSE}_k^{AnF}$, is reported

in Table 1. The maximum selection probability obtained with the AnF is $57\%$, which is smaller compared to the experiments using AnDA ($64\%$ using RMSE of the forecast error and $68\%$ using CME). This stresses the importance of having an accurate initial conditions in order to be able to compare the forecasts obtained with two competing models.

### 3.2 SPEEDY model

In this section, we discuss the implementation of CME with AnDA for the Simplified Parameterizations, primitivE-Equation

DYnamics (SPEEDY; Molteni, 2003) which is an intermediate complexity, atmospheric general circulation model. The grid in SPEEDY consists of 48 points in the South-North direction and 96 points in the West-East direction and of 7 vertical $\sigma$-levels. SPEEDY has a set of simplified parameterizations to represent unresolved scale processes including radiation, large



| Approaches | | $T = 10^2$ | $T = 10^3$ | $T = 10^4$ |
|---|---|---|---|---|
| EnKF | Mean (%) | 68.64 (64.37) | | |
| | CIs (%) | [67.59, 69.70] ([63.59, 65.25]) | | |
| AnEnKF | Mean (%) | 59.85 (58.99) | 67.02 (63.33) | 68.44 (64.37) |
| | CIs (%) | [58.74, 60.96] ([58.00, 59.99]) | [65.43, 68.60] ([61.92, 64.73]) | [67.09, 69.79] ([63.59, 65.25]) |
| AnF | Mean (%) | $-$ (53.38) | $-$ (57.46) | $-$ (57.71) |
| | CIs (%) | [$-$, $-$] ([52.11, 54.60]) | [$-$, $-$] ([56.61, 58.30]) | [$-$, $-$] ([56.79, 58.63]) |

**Table 1.** Sensitivity of the percentage of correct model identification based on the CME and on the RMSE (values within parenthesis) with respect to the length of the catalog ($T$) used in the AnEnKF and in the AnF. The mean and $95\%$ confidence interval of the percentage of correct model identification are computed for each of the algorithms using 10 repetitions of the assimilation experiment.

scale condensation, soil-sea-ice-atmosphere energy fluxes, boundary layer, and moist convection. A brief description of these schemes can be found in the Appendix of Molteni (2003).

### 3.2.1 Data driven model selection with SPEEDY

In this work, we conduct observation simulation experiments using this model. A 30-year run is performed using the default configuration of the model which has been shown to produce a good representation of the main features of the current climate (Molteni, 2003). The model is integrated with a time step of 40 minutes and model states are archived at 6-hours intervals. This simulation is hereafter referred to as the TRUE simulation.

To simulate imperfections in the model formulation, we modify the value of the $RH_{cnv}$ parameter which is related to the deep convection parameterization in SPEEDY (Molteni, 2003). This parameter controls the activation of the convective parameterization and the intensity of the convective overturning. Lower values of the parameter lead to more frequent convection activation and stronger vertical mass fluxes. The TRUE simulation uses $RH_{cnv} = 0.9$. Two additional simulations are performed using $RH_{cnv} = 0.8$ and $RH_{cnv} = 0.7$ which are referred to as $RH08$ and $RH07$ respectively. Figure 5 shows that reducing $RH_{cnv}$ leads to an increase in mid-level mean temperature within in mid-latitudes and in the tropics (with some exceptions on the Western Pacific and Northern Africa). This increase is related to stronger latent heat release within the intertropical convergence zone (ITCZ, Figures 5 e and f ) and to enhanced subsidence in the polar side of the Hadley circulation. Precipitation produced by the convective scheme is enhanced over tropical regions which can also contribute to the mid-level warming in this region. $RH07$ and $RH08$ produce 12% and 6% more convective precipitation than the TRUE experiment respectively. However, the sensitivity in total precipitation is much smaller due to a decrease in the large scale precipitation as the convective precipitation increases. The impact of this parameter upon the mean distribution of temperature is somehow linear, since both $RH08$ and $RH07$ produce anomalies with a similar spatial pattern but larger amplitude for the later.

The AnDA approach in SPEEDY is implemented over local domains centered at each model grid point. In this work and for each grid point in the horizontal domain, analogs are defined based on the value of the temperature on a 3-dimensional box





of 3 by 3 horizontal grid points and 3 vertical $\sigma$-levels ($\sigma$=0.77, 0.6, and 0.42). Temperature is selected since its horizontal distribution and its vertical gradient are directly affected by convective processes, particularly in the tropics, as has been shown in Figure 5.

Temperature observations are generated from the TRUE run at each model grid point adding uncorrelated Gaussian random errors with a standard deviation of 0.7 K. The assumed error standard deviation is similar to the assumed error of several real
temperature observations provided by radiosondes and satellite retrievals. Note that data assimilation cycles performed for each grid point are completely independent of their neighbors. Although, in these experiments, the computation is performed globally, results would be the same if the method was applied regionally. This local implementation resembles the local computation of the CME introduced by Metref et al. (2019). However, the method implemented here is different since the analysis cycle at a particular grid point is performed using only the observations within the local domain. Moreover, by implementing
AnDA locally we significantly increase the probability of finding relevant analogs reducing the size of the catalog required for an accurate approximation of the dynamical model. Also, the local implementation allow us to avoid the global integration of the model resulting in a substantial reduction of the computational cost.

AnDA experiments are conducted assimilating the observations generated from the last three years of the TRUE simulation. The catalogs for the analog forecasting are constructed from the first 25 years of the $RH08$, $RH07$, and TRUE model runs and
250 analogs are used for the forecast. The number of ensemble members is 30. The assimilation frequency is set to 24 hours and to take advantage of 6-hourly data, 4 data assimilation cycles staggered in time are performed at each grid point. The analysis obtained from these experiments are merged to obtain a total of 4,380 analysis cycles over the three-years assimilation period.

As in the Lorenz-63 experiments, the adaptive multiplicative inflation method indicated in Alg. 1 is used to find the optimal
inflation value corresponding to each experiment. Figure 6 shows that CME is quite sensitive to the multiplicative inflation used in AnDA. As expected, the optimal inflation value (the one which produce the maximum CME) is larger for the catalogs corresponding to the imperfect models. Moreover, if a large inflation is assumed (e.g., larger than $1.5$), CME associated with $RH08$ becomes larger than the one obtained with perfect model catalog. Also, the adaptive inflation produces results which are close to the ones obtained with the optimal fixed multiplicative inflation value thus avoiding the need to manually tune
the inflation parameter. This is important, considering the results of Miyoshi (2011) which show that for atmospheric general circulation models, the optimal inflation parameter depends on time and location.

Preliminary experiments performed to optimize the configuration, show that results are particularly sensitive to the assimilation frequency and to the size of the local domain used to identify analogs. In particular using 3 different vertical levels and less frequent assimilation, results in a much stronger sensitivity to $RH_{cnv}$ (i.e. larger difference in CME associated with different
catalogs). The number of analogs affects the performance of AnDA but has a lower impact on the relative performance of the different models.

First, it was checked that the proposed methodology is able to identify that the temperature observations were generated using the TRUE simulation and not the $RH08$ or $RH07$ runs. To evaluate the statistical robustness of the results, all the experiments were repeated 10 times using different realizations for the observation noise. The standard deviation of the values





obtained from different experiments is used to estimate the $95\%$ confidence interval for the different metrics discussed in this section.

As described by Metref et al. (2019), model comparison in the context of data assimilation can be conducted using different metrics. We compare the performance of the different models using RMSE of the forecast error in the observation space ($\mathrm{RMSE}^f$) and CME. Figures 7 (a) and (b) show the percentage difference in $\mathrm{RMSE}^f$ and CME for the experiments with $RH08$

and $RH07$ with respect to the one using the TRUE catalog. The percentage difference is computed as the score difference with respect to the score corresponding to the TRUE catalog divided by the score obtained with the TRUE catalog (e.g. the percentage difference for the RMSE and for the RH07 experiment is obtained as: $(\mathrm{RMSE}^f_{RH07} - \mathrm{RMSE}^f_{TRUE})/\mathrm{RMSE}^f_{TRUE}$). The performances of the experiments using catalogs $RH08$ and $RH07$ are consistently worse than the one of the experiments using the TRUE catalog (positive $\mathrm{RMSE}^f$ percentage differences). Larger differences are found in the tropics where convection

is more frequent and stronger. This suggests that AnDA provides a valuable hint about the source of model imperfection.

Figures 7 (c) and (d) show the percentage difference for CME. In the case of CME, negative percentage differences indicate a worse performance with respect to the experiment that uses the TRUE catalog. The CME has a similar spatial pattern as the difference in RMSE. However, for the CME, the area where results are statistically significant is larger (see for example Asia, Europe and Hawaii for the experiment $RH07$). Figure 8 shows that the spatial pattern of the percentage of correct

identification computed from RMSE or CME is similar. However, the CME generally identifies the right model more frequently and the number of grid points at which the percentage of correct identifications is significantly over 50% is larger. This is mainly because CME incorporates the uncertainty in the quantification of the fit of the forecast to the observations. In these experiments, an accurate estimation of this uncertainty is achieved combining the ensemble of analog forecasts and the adaptive multiplicative inflation. A large error will contribute more to the CME if the estimated forecast uncertainty is low than if it is

large. This helps to reduce the effect of stochastic errors in the initial conditions and its amplification due to the chaotic nature of the system.

To quantify the advantage of using data assimilation in identifying the catalog that best fits the observations, we compare AnDA results with an experiment using the AnF approach (i.e., in which the analog forecast is initialized directly from the noisy observations and evaluated using the $\mathrm{RMSE}^{AnF}$). During the same period corresponding to the data assimilation experiment,

24-hour analog forecast are initialized every 6 hours using the same three catalogs as in the AnDA experiments. The $\mathrm{RMSE}^{AnF}$ is used to compare the performance of the different catalogs. Figure 7 (e) and (f) show that differences in performance are mixed when no data assimilation is used. Larger RMSEs are associated with the catalogs generated with imperfect models at mid-latitudes. However, over the tropic the opposite is observed with the imperfect catalogs producing better forecasts. Moreover the area in which differences between the imperfect model catalogs and the TRUE is larger in the experiments that use AnDA.

These results highlight again the importance of using DA to quantify the performance of a dynamical model based on noisy observations. When DA is not used, the forecast associated to a model may be corrupted by a large initialization error and the resulting $\mathrm{RMSE}^{AnF}$ be sensitive not only to the model error but also to its sensitivity to initial conditions.





### 3.2.2 Standardized catalog experiments

To evaluate if the CME framework can detect differences in the system' dynamics that go beyond the change in the probability
distribution of the state variables (i.e. their climatological mean and standard deviation) we evaluate the skill of AnDA to
detect differences in the model performance when the differences in the mean and in the standard deviation among the catalogs
is removed. To achieve this we perform a standardization of the state variables and the true state prior to the generation of
the observations. The seasonal cycle is taken into account in the standardization using a 60-day centered moving window to
compute the climatological mean and standard deviation corresponding to different days of the year. In the AnDA experiments
performed using standardized catalogs and observations, the observation error has been scaled accordingly. Results of the
percentage of correct identification for both RMSE$^f$ and CME are shown in Figure 9. Differences in RMSE$^f$ and CME among
different catalogs are lower than in previous experiments. This indicates that the difference in the mean and standard deviation
among the catalogs contributes to CME. However, when only standardized data is used, AnDA is still able to identify the
correct model with the higher probabilities obtained by using the CME. RMSE$^f$ shows correct model selection probabilities
over the tropical regions where the signal is stronger, however, there are some areas over mid-latitudes where the percentage
of correct identification is consistently below 50% resulting in mixed results for this metric. For some of the areas where the
RMSE provides the wrong answer, the CME is still able to provide percentages over 50% (see for example the Northern and
Southern Pacific).

Figure 10 (a) and (b) shows the percentage of correct identification based on CME using an evidencing window of h=7 days.
This is higher than the one obtained using a single-day (h=1) window (as in Figure 8 (c) and (d)). For the 7-days window,
the percentage of correct identification is well above 70% over a large part of the world suggesting that the impact of model
imperfection on climatological events occurring at time-scales in the order of weeks can be correctly detected most of the
time. Also, the signal obtained when the standardized anomalies are assimilated is more clear when a 7-days window is used
(Compare Figures 10 (c) and (d) with Figures Figure 9 (c) and (d)).

### 3.2.3 Seasonal dependent model errors

To evaluate if the CME computed using AnDA can capture the temporal variability of model errors the percentage of correct
identification is computed over two sub-periods, one corresponding to the Northern Hemisphere summer (June, July, and
August) and the other one corresponding to the Northern Hemisphere winter (December, January, and February). Figure 11
shows that the sensitivity in the mid-level mean temperature and mean precipitation in these two periods is quite different,
with some areas showing the opposite sensitivity (e.g. central South America, Southern North America, Southern Africa and
Australia).

Figure 12 shows that the CME for both seasons is quite different and that the evidence is generally stronger in the summer
hemisphere when convection is more frequent. Note in particular that over the North Atlantic and Europe and Asia, the CME is
stronger during the summer. According to the SPEEDY model climatology convective rain is larger during this time of the year
(not shown) and so is the sensitivity of this precipitation to the $RH_{conv}$ parameter (see Figure 11 (c) and (d)). The seasonal





cycle is acting as a state-dependent source of model error and the CME successfully recovered this characteristic providing useful information about the source of the difference among the models used to generate the different catalogs.

## 4    Discussion and conclusions

Model selection using data assimilation has been introduced in Carrassi et al. (2017), applied in the context of climate change
detection and attribution in Hannart et al. (2016), and to complex or large state model evaluation in Metref et al. (2019). It consists in putting in competition two or more dynamical models and use observations to compute a likelihood, also called CME, for each model, attributing the highest probability to the model which provides forecasts that better match the observations over a period of time. The CME compares the observations with the mean of the forecast distribution computed in the assimilation cycles, but also takes into account the uncertainties described by the variance of the forecast distribution.

The main issue related to model selection using data assimilation is the computational cost to perform several model runs/evaluations. Recently, a data-driven data assimilation method based on analog forecasting has been proposed in Tandeo et al. (2015) and Lguensat et al. (2017). It consists in replacing the physical model equations by a catalog of past observations or numerical simulations to statistically emulate the dynamics of the system. This current paper explores the use of such a model-free strategy in data assimilation to compute the CME of several dynamical models, represented by a set of numerical
simulations.

The proposed methodology was assessed using numerical experiments. A first set of experiment was performed using a modified three-variable Lorenz-63 model. It was found in particular that using analog data assimilation gives similar results than with classical data assimilation, where the dynamical model is run at each time step. It indicates that the method is able to provide an accurate approximation of the CME metric of a dynamical model given a catalog and set of noisy observations. A
second set of experiment was done using the intermediate complexity atmospheric general circulation model SPEEDY. They indicate that the proposed methodology is efficient in identifying the correct model using only observations of a small part of the state. The numerical results also highlight the importance of using data assimilation compared to more direct approaches which rely only on the forecast sensitivity without proper state initialization. As a summary, numerical results indicate that this technique is efficient in selecting the best model to describe a sequence of noisy observations, and it has various advantages:
(i) it is a low computational cost method, (ii) it can be applied locally on a sub-part of the system or using complex observation operators such as integrated parameters, (iii) it uses already existing model outputs.

This work is a first step of a more challenging project. It shows that selecting and weighing dynamical models can be performed inside a data assimilation framework, using analog forecasts, and thus avoiding the need to run numerical models to get predictions. This result opens new perspectives for the use, for instance, of model simulations such as the Coupled Model
Intercomparison Project (CMIP), see Eyring et al. (2016) for more details. The goal will be to propose weighted projections of climate indices, where the weights will be based on the skill of different climate models on representing the local dynamics and the current observations. The resulting weighted projections will improve the estimation of the mean and standard deviation



of climate indices. Those results will be compared to model democracy strategy, for instance, which gives equal weights to all climate models, but is also somehow controversial (Knutti et al., 2019).

*Code and data availability.* Codes can be obtained from the GitHub open repository: https://github.com/gustfrontar/AnDA_SPEEDY.git, https://doi.org/10.5281/zenodo.5803356 (Ruiz and Tandeo, 2021). They include the codes for aproximating the SPEEDY model using the analog forecasting technique, some simulated data and the specific version of AnDA used for the SPEEDY experiments. AnDA is an open-code system and can be obtained from the GitHub open repository: https://github.com/ptandeo/AnDA, https://doi.org/10.5281/zenodo.5795943 (Tandeo and Navaro, 2021).

*Author contributions.* All authors contributed in the model development, verification, discussions about the results, and preparation of the manuscript. TTTC and JR were responsible for the numerical experiments and prepared the figures.

*Competing interests.* The contact author has declared that neither they nor their co-authors have any competing interests.

*Acknowledgements.* We thank the ECOS-Sud Program for its financial support through the project A17A08. We also thank financial support for the AMIGOS project from the *Région Bretagne* (ARED fellowship for PLB) and from the ISblue project (Interdisciplinary graduate 415 school for the blue planet, ANR-17-EURE-0015) co-funded by a grant from the French government under the program *Investissements d'Avenir*. This research has also been supported by the National Agency for the Promotion of Science and Technology of Argentina (grant no. PICT-2033-2017), the University of Buenos Aires (grant no. UBACyT-20020170100504).



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



**Figure 1.** Schematic representation of the proposed methodology. The procedure is iterative, from an initial to a final time index. At time index $k-1$, the procedure starts with the results of different data assimilation systems in (a), corresponding to different dynamical models. In (b), each analysis state is used to find the nearest analogs. Those analogs and corresponding successors, coming from catalogs of numerical simulations, are used to build analog regressions. The resulting probabilistic forecasts given in (c) are compared to the available observations at time index $k$. Then, the likelihoods (CMEs) are used to compute a weight for each dynamical model.



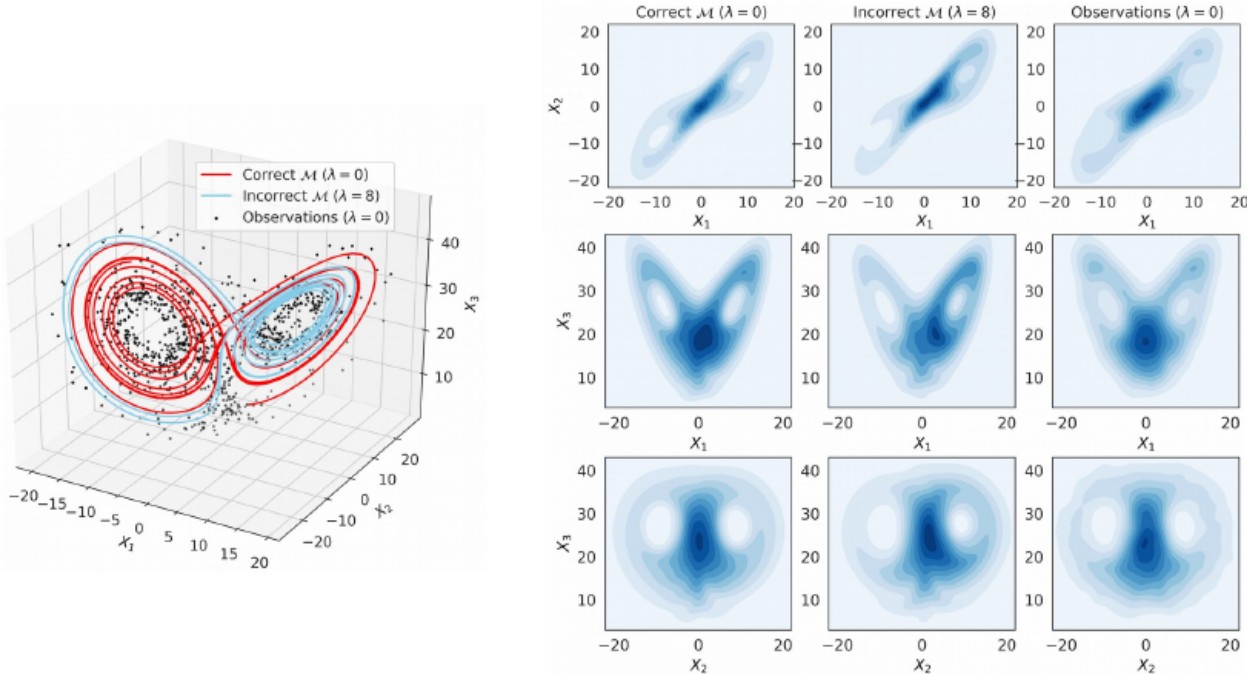

**Figure 2.** Simulated trajectories (left) and bivariate distributions (right) from the modified Lorenz-63 model Eq. (16) with $\lambda = 0$ and $\lambda = 8$. The observations (dots) are generated with the correct model with $\lambda = 0$ and an additive Gaussian noise with mean $0$ and variance $2I_3$. Figure obtained using a time step $\mathrm{d}t = 0.01$.



**Figure 3.** Time averaged CME as a function of the parameter $\lambda$ used to generate the incorrect model for the EnKF (left) and AnEnKF (right) approaches using different number of members $N$ (rows). Adaptive inflation is used only for the results in the bottom row. The red line indicates the time averaged CME for the correct model ($\lambda = 0$).

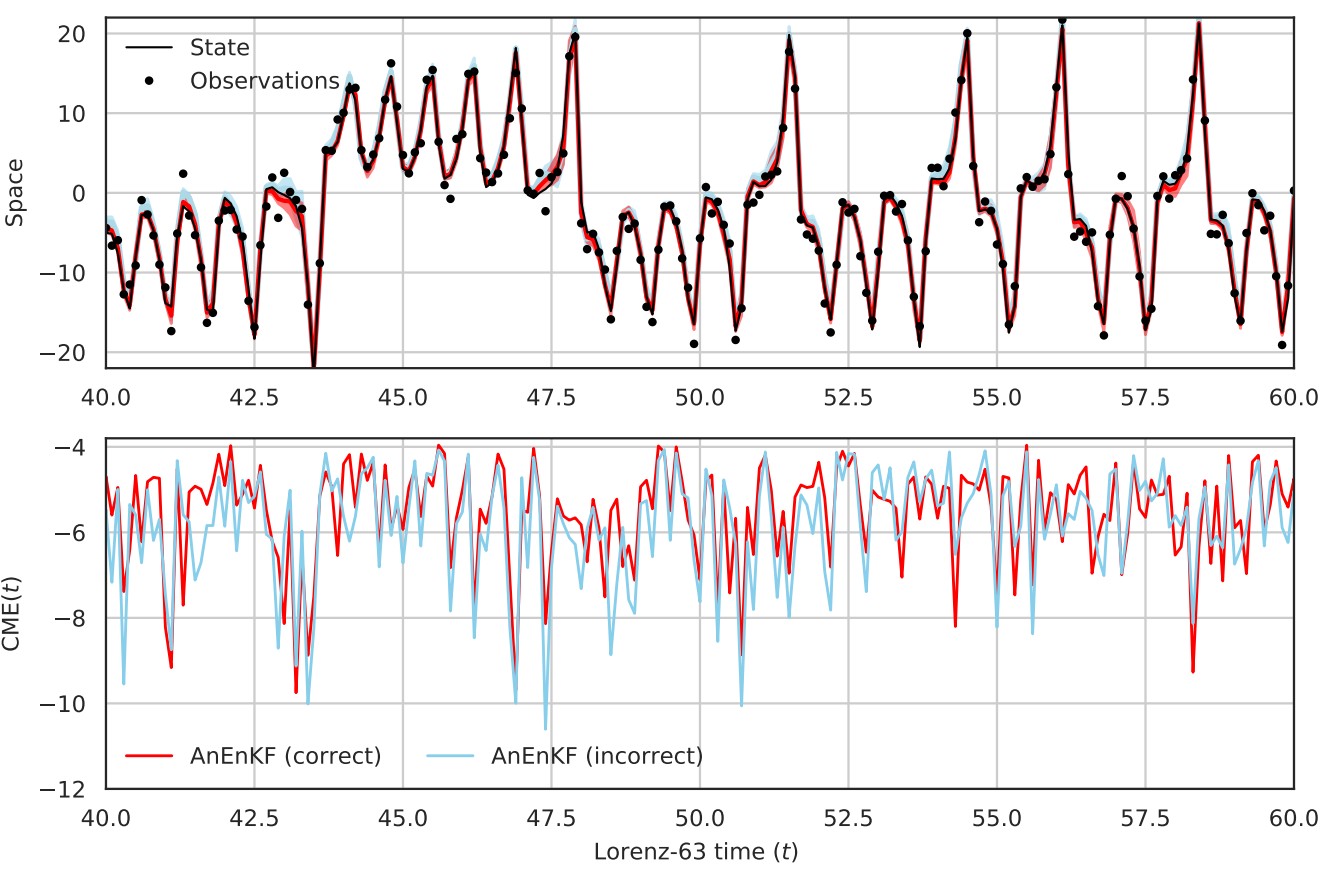

**Figure 4.** Top panel : time series of the second component of the Lorenz-63 system with mean analysis and $95\%$ prediction interval obtained using AnEnKF with the correct $\mathcal{M}^{(0)}$ model (red) or using AnEnKF with the incorrect $\mathcal{M}^{(8)}$ model (blue). Bottom panel : CME time series for the correct $\mathcal{M}^{(0)}$ model and the incorrect $\mathcal{M}^{(8)}$ model. $N = 20$ members are used and inflation factor is applied in AnEnKF.

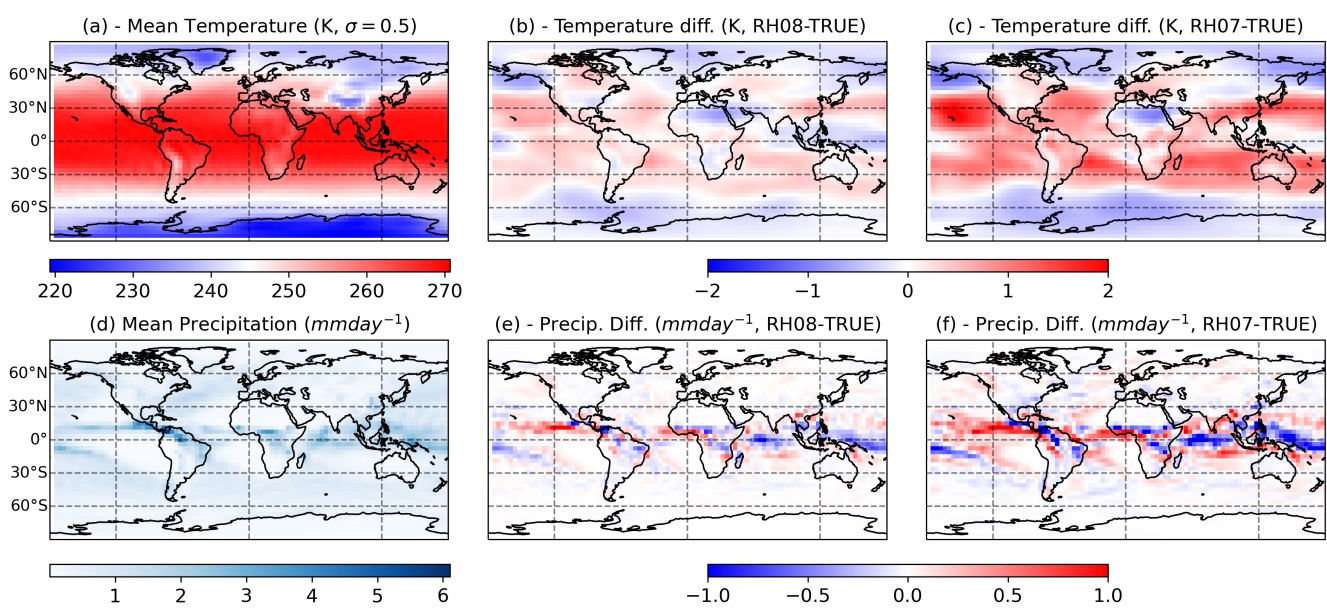

**Figure 5.** 30-year mean for the TRUE model simulation (left column) and differences between the TRUE model simulation and the $RH08$ and $RH07$ mode simulation (center and right column, respectively), for temperature at the vertical level $\sigma = 0.5$ (top row, $K$) and total precipitation (lower row, $mmday^{-1}$).





**Figure 6.** Mean CME for a grid points located over Northern South America as a function of the inflation coefficient. The shade shows the range between the maximum and minimum values over 10 repetitions of the experiment. The error bars represents the range between the maximum and minimum values of CME and estimated inflation over 10 repetitions of the experiment. Results are shown for the TRUE (blue line), $RH08$ (green line), and $RH07$ (red line) model experiments.







**Figure 7.** Percentage difference in RMSE$^f$ (first row) and CME (second row) for the AnDA experiments performed using the $RH07$ and $RH08$ catalogs (left and right column, respectively) versus the experiment performed with the TRUE catalog. The third row correspond to the difference in RMSE$^{AnF}$ from the experiment based only on analog forecasting (AnF). In all panels, the gray shade indicate values which are below the 95% statistical significance level.

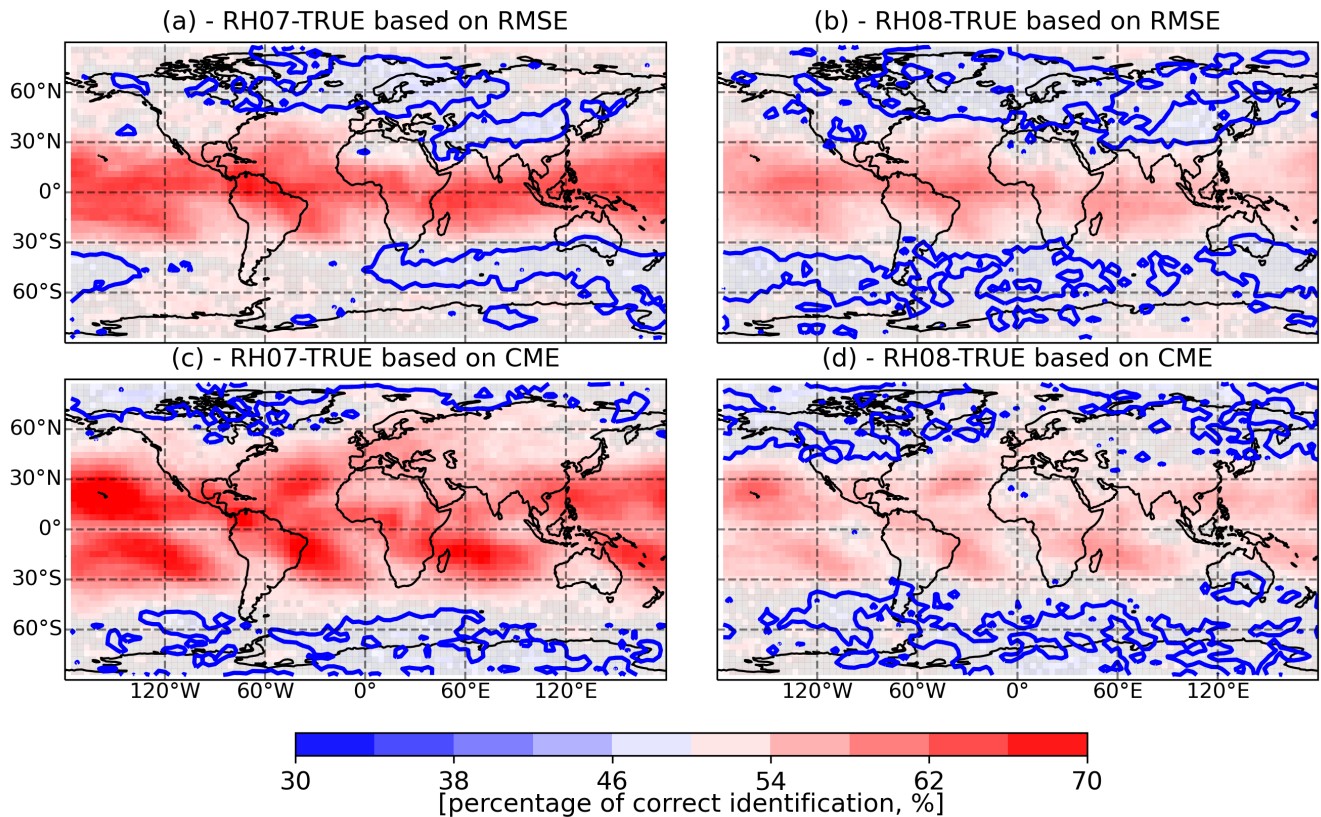

**Figure 8.** Percentage of correct identification between the $RH07$ and TRUE catalog (left column) and between the $RH08$ and TRUE catalogs (right column) using AnDA. The percentage of correct identification is computed using either $RMSE^f$ or CME (upper and lower row, respectively). The gray shade indicate values which are below the 95% statistical significance level. The blue line corresponds to a 50% of correct identifications.



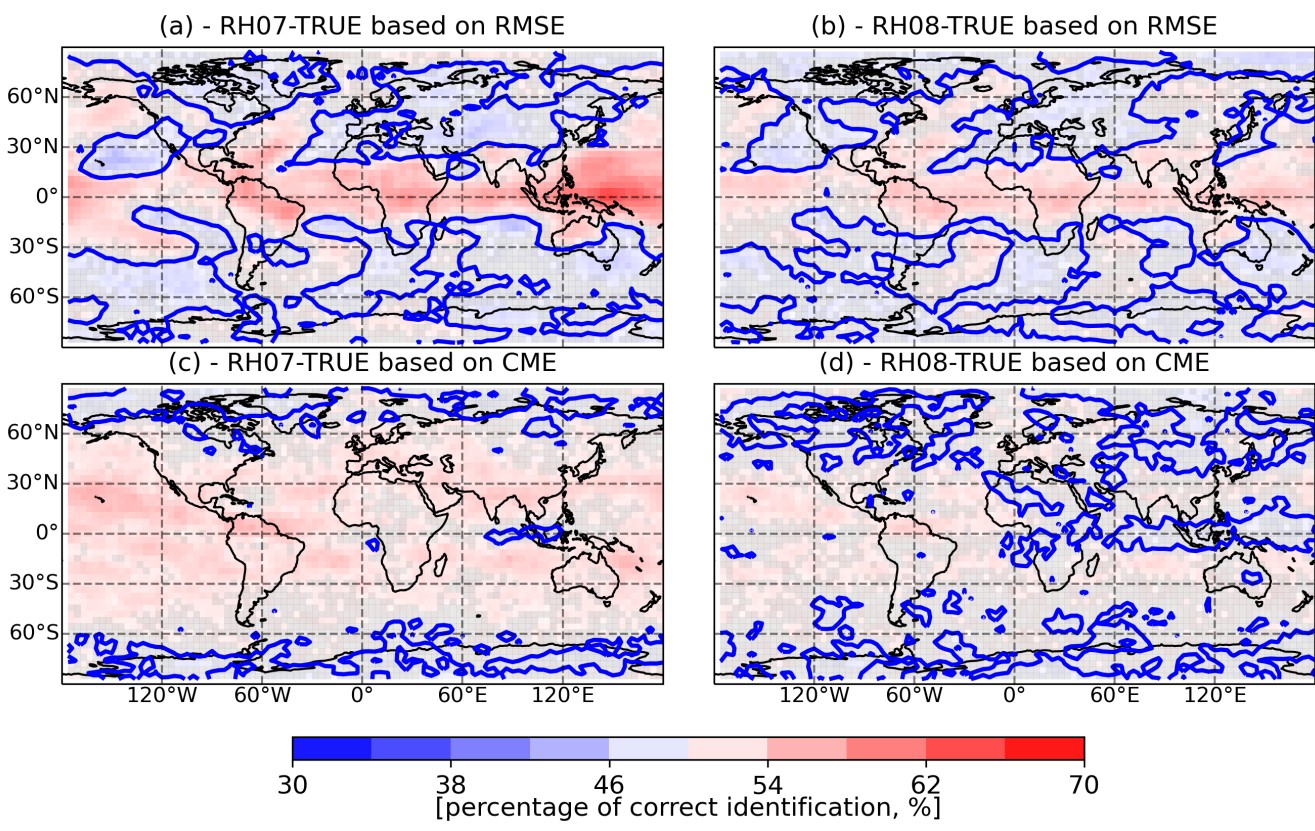

**Figure 9.** As in Figure 8 but for the experiments in which the catalogs and observations have been standardized with respect of their respective climatological variability (see the text for details).



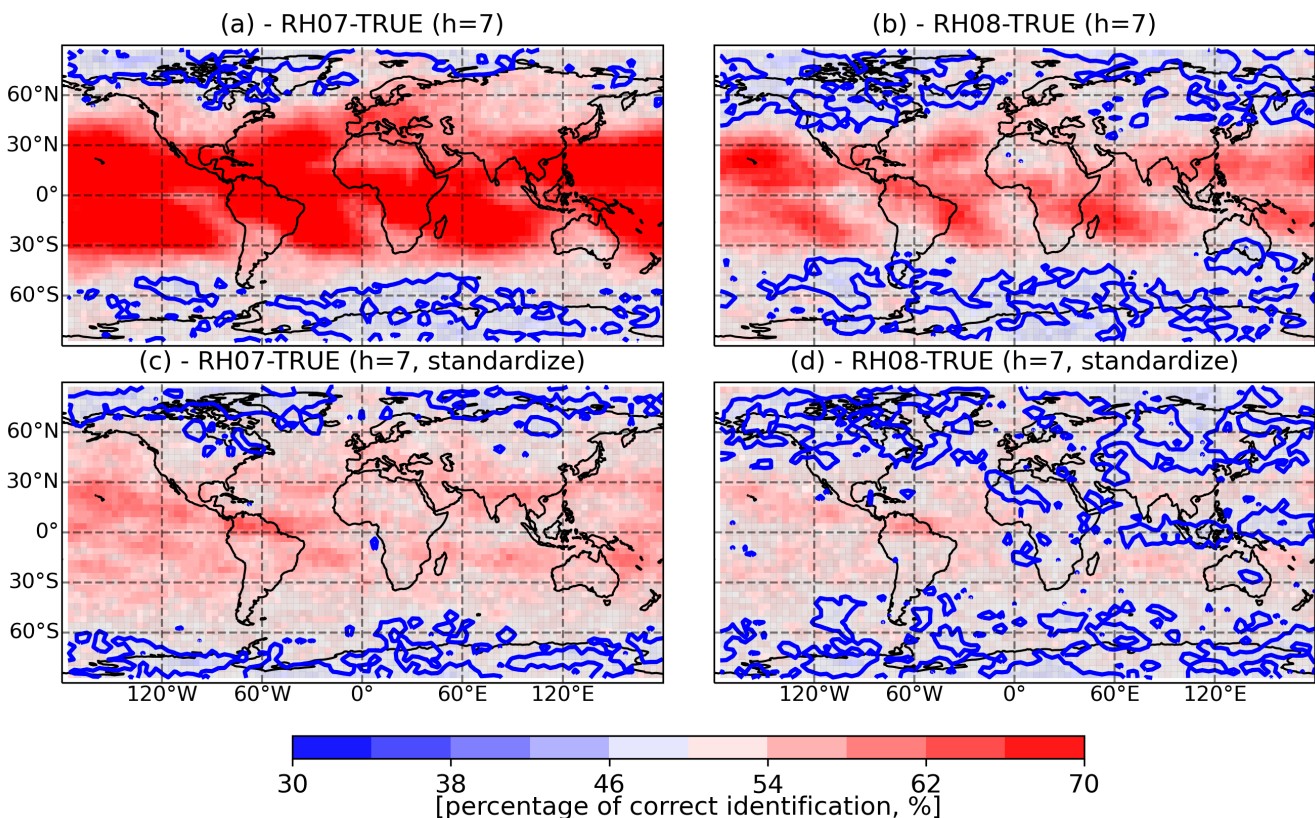

**Figure 10.** Percentage of correct identification based on the CME for a 7-days evidence window between the $RH07$ and TRUE catalog (left column) and between the $RH08$ and TRUE catalogs (right column) using AnDA. The percentage of correct identification is computed assimilating temperature observations or their standardized values (upper and lower row, respectively). The gray shade indicate values which are below the 95% statistical significance level. The blue line corresponds to a 50% of correct identifications.

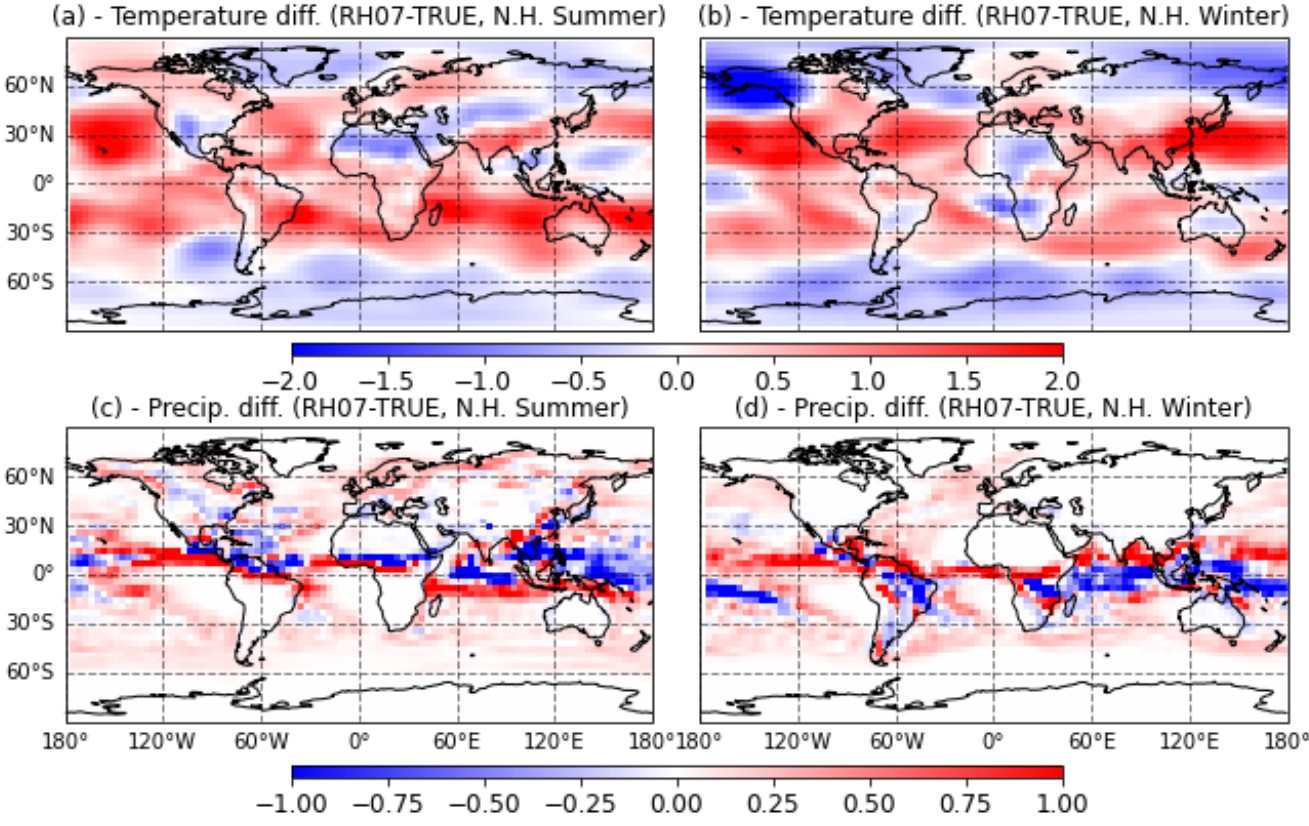

**Figure 11.** 30-year mean difference between the TRUE model simulation and the $RH07$ for the Northern Hemisphere Summer (June, July, and August, left column) and the Northern Hemisphere Winter (December, January, and February, right column) for temperature at $\sigma = 0.5$ (top row, $K$) and for convective precipitation (bottom row, $mmday^{-1}$).



**Figure 12.** Percentage of correct identification between $RH07$ and TRUE catalog (left column) and between the $RH08$ and TRUE catalogs (right column) based on CME for the Northern Hemisphere winter (December, January, and February, first row), the Northern Hemisphere summer (June, July, and August, second row). The third row shows the difference between Northern Hemisphere winter minus summer in the percentage of correct identification. The gray shade indicate values which are below the 95% statistical significance level. The blue line corresponds to a 50% of correct identifications.