# Peer review of "Analog Data Assimilation for the Selection of Suitable General Circulation Models"

_Geoscientific Model Development, 2021_

## Referee Comment (RC1)

**GMD-2021-434 Authors: Juan Ruiz, Pierre Alliot, Thi Tuyet Trang Chau, Pierre Le Bras, Valerie Monbet, Florian Sevellec, and Pierre Tandeo**

11 March, 2022

Title: *Analog Data Assimilation for the Selection of Suitable General Circulation Models*

Recommendation: Accept subject to **Minor Revision**

Dear Editor,

this is a very good and original work pushing forward a research line that was initiated only a few years ago. In particular this study investigates the use of a model-free data assimilation framework to perform model selection. The key novelty of this work stands on the use of the "analog" method to replace the very costly forward model computation.

I am very favourable about accepting the manuscript. I have however some minor points that I would ask the Authors to address before acceptance. Specifically about the lack of sufficient clarity on the details of the proposed approach, and on readability.

My specific minor points follow below:

1. Given that you defined the acronym DA, I would encourage the Authors to use it throughout the manuscript. In many instances the full wording is used instead.

2. Line 9–10. The sentence is not sufficiently clear to be in the Abstract.

3. Line 37. Maybe better "selection" in the place of "evaluation".

4. Line 51. The citation should go in between parentheses.

5. Equation 1. The CME is given by the sum of the individual CMEs. You must recall that the system must be Markovian and the observations independent [Carrassi et al., 2017].

6. Equation 6. The model error term, $\eta$ should be bold face. Is it a vector, isn't?

7. Line 104. Please specify which $\mathbf{Q}$ did you use? Is this the same for both the EnKF and the AnEnKF? Is this the same for all value of $\lambda$? Also, I think you should use the bold face notation.

8. Equation 13. I acknowledge the Authors use of inflation, but I wonder: are you also using localisation in the experiments with SPEEDY?

9. Line 147. The citations are in the text and should go without parentheses.

10. Line 148. "the the".

11. Line 149. "An interesting feature of AnDA is that it can be applied locally, without the need to approximate the full model." This sentence, in particular with regards to the locality, must be clarified.

12. Line 159. "Note that this regression is nonlinear since the linear approximation is applied locally in state space and corresponds to a first-order expansion of the dynamical model (see Platzer et al., 2021).

13. Equation 15. Are you saying that the $N$ realisations of Eq. 6 are substituted with the single realisation in Eq. 15? How do you mimic/get ensemble spread then? Please clarify.

14. Line 182. "have" should be "has".

15. Line 183. "Secondly, model errors associated with the local linear regression are considered and contribute to the forecast ensemble spread." Very unclear, please explain better.

16. Line 204–205. What is the order of the Runge-Kutta? It is written "4-5". Then, is the model time step, $\mathrm{d}t = 0.1$, equals to the assimilation interval?

17. Line 215–216. "Figure 3 shows the mean CME for different values of $\lambda$". I think this sentence can be moved before when you started describing the figure.

18. Line 221. $x$ and $x^a$ should be bold.

19. Caption of Table 1. It should be "parentheses".

20. Line 267. Do you mean "latter"?

21. Lines 275–276. "Note that data assimilation cycles performed for each grid point are completely independent of their neighbours". This is very unclear. Please explain better, in particular about the concept of gridpoints independence that seems contradicting the multivariate nature of DA.

22. Lines 277–282. This part is also very unclear. You need to work it out better as it contains key details of your method. For instance, you need to construct your catalogue by running the full model anyhow, isn't? Then I understand you search for analog within the 3x3 domain.

23. Line 285. Please explain from where the number of 250 analogs comes from the other experiments details.

24. Line 286. What do you mean by "staggered"? What is the difference then between assimilation and observation frequency?

25. Line 287. Please explain how do you get 4,380 analysis cycles.

26. Line 312. I suggest that you include the equation as a formula in a generic way valid for RMSE and CME.

27. Lines 324–326. I overall agree of what is said right before and it seems to me that these conclusions are in line with those Metref et al, 2019 and Carrassi et al, 2017. However, I do not understand the lines here: what large errors are you talking about?

28. Line 397. Do you mean "weighting"?

29. Conclusion. I think the Authors results is also potentially very relevant for the so called "super-modelling" approach, whereby a dynamic combination of imperfect models is built. There the issue on how to "weight" the different models entering the combination is very important (see e.g. *Schevenhoven et al.* (2019)) and their approach may provide a ranking of model quality.

**References**

Schevenhoven, F., F. Selten, A. Carrassi, and N. Keenlyside, Improving weather and climate predictions by training of supermodels, *Earth System Dynamics*, *10*, 789–807, 2019.

---

## Author Comment (AC1)

**Reviewer 1**

GMD-2021-434 Authors: Juan Ruiz, Pierre Alliot, Thi Tuyet Trang Chau, Pierre Le Bras, Valerie Monbet, Florian Sevellec, and Pierre Tandeo
11 March, 2022
Title: Analog Data Assimilation for the Selection of Suitable General Circulation Models
Recommendation: Accept subject to Minor Revision
Dear Editor, this is a very good and original work pushing forward a research line that was initiated only a few years ago. In particular this study investigates the use of a model-free data assimilation framework to perform model selection. The key novelty of this work stands on the use of the "analog" method to replace the very costly forward model computation.
I am very favourable about accepting the manuscript. I have however some minor points that I would ask the Authors to address before acceptance. Specifically about the lack of sufficient clarity on the details of the proposed approach, and on readability.

We would like to thank the reviewer for many comments that improved the discussion of the results, the clarity and many presentation aspects of the manuscript.

My specific minor points follow below:

1. Given that you defined the acronym DA, I would encourage the Authors to use it throughout the manuscript. In many instances the full wording is used instead.

We agree with the reviewer. We used the DA acronym as much as possible except in the last section which is sometimes read separately from the rest of the text.

2. Line 9–10. The sentence is not sufficiently clear to be in the Abstract.

We agree with the reviewer. The sentence is not clear enough. We propose to replace the following sentence: "Moreover, the technique is sensitive to differences in the model dynamics which are not reflected in the moments of the climatological probability distribution of the state variables." by "Moreover, the technique is able to detect differences among models in terms of local dynamics in both time and space which are not reflected in the first two moments of the climatological probability distribution".

3. Line 37. Maybe better "selection" in the place of "evaluation".

We followed the reviewer's suggestion

4. Line 51. The citation should go in between parentheses.

We corrected this as proposed by the reviewer.

5. Equation 1. The CME is given by the sum of the individual CMEs. You must recall that the system must be Markovian and the observations independent [Carrassi et al., 2017].

Thank you very much for pointing this out. We modified the introduction of Equation 1 as follows: Remark that as shown by Carrassi et al. 2017, for a Markovian system and for independent observations

$$\mathrm{CME}_{k:k+h}(\mathcal{M}) = \sum_{i=k}^{k+h} \ln p\left(\mathbf{y}_i | \mathbf{y}_{0:i-1}; \mathcal{M}\right) = \sum_{i=k}^{k+h} \mathrm{CME}_i(\mathcal{M})$$

6. Equation 6. The model error term, η should be bold face. Is it a vector, isn't?

Thank you very much for spotting this. We corrected equation 6.

7. Line 104. Please specify which Q did you use? Is this the same for both the EnKF and the AnEnKF? Is this the same for all value of λ? Also, I think you should use the bold face notation.

In line 104 (of the original manuscript), Q denotes the covariance of model noise in general discrete dynamical models. If Q is known (in experiments), we can use the true value of Q in both EnKF and AnEnKF. If Q is unknown (in reality), we can use its estimates. The estimates are usually derived from likelihood-based or Bayesian methods for EnKF and from analog forecast residuals for AnEnKF.
In L63 experiments, we did not consider a forecast noise in the simulations (Q=0). The results of EnKF (Q=0) have been used as a benchmark to assess the results of AnEnKF. We use this same configuration for all experiments independently of the value of λ. We add the following statement to clarify this point when we describe the experiments performed with the Lorenz 63 model:
"For the EnKF, experiments with **Q**=0 are used independently of the value of λ, while in the AnEnKF the approach described in the previous section is used, which in practice corresponds to a robust estimate of the model noise."
We also corrected the notation of **Q**, it is now in boldface. Thank you.

8. Equation 13. I acknowledge the Authors use of inflation, but I wonder: are you also using localisation in the experiments with SPEEDY?

We thank the reviewer for raising this point. We add the following explanation in the description of the SPEEDY experiments:

"In ensemble-based DA methods, usually the estimation of the forecast error covariance from a limited-size ensemble leads to sampling noise. This is usually ameliorated by the use of localization schemes that reduces the amplitude of the covariance between distant variables. In this paper AnDA is implemented without localization given the relatively small size of the local domain in which DA is performed."

9. Line 147. The citations are in the text and should go without parentheses.

Thank you very much. We corrected this in the revised version of the manuscript.

10. Line 148. "the the".

Thank you very much. We corrected this in the revised version of the manuscript.

11. Line 149. "An interesting feature of AnDA is that it can be applied locally, without the need to approximate the full model." This sentence, in particular with regards to the locality, must be clarified.

We agree with the reviewer in which additional clarification was required at this point. We propose rewriting that part of the text as follows:

"It is a flexible framework that can be adapted to a large set of problems. An interesting feature of AnDA is that it can be applied locally, without the need to approximate the full model meaning. On the contrary, only a part of the state space (e.g. a particular region or physical variable) can be used to select the analogs and to emulate the dynamics of that particular part of the system. This is particularly advantageous when dealing with high dimensional systems."

12. Line 159. "Note that this regression is nonlinear since the linear approximation is applied locally in state space and corresponds to a first-order expansion of the dynamical model (see Platzer et al., 2021).

We could not follow this particular comment. Please provide us with more details about this particular comment. At the same time we notice that this sentence is not clear enough so we rephrased it in the following way:
"Note that this regression is able to emulate the model dynamics including non-linearities as it is a linear approximation applied locally in state space and corresponds to a first-order expansion of the dynamical model (see Platzer et al., 2021)"

13. Equation 15. Are you saying that the N realisations of Eq. 6 are substituted with the single realisation in Eq. 15? How do you mimic/get ensemble spread then? Please clarify.

We agree with the reviewer. The equation was not consistent with what has been previously stated. We include explicitly the ensemble member index in Equation 15 to clearly state that we are propagating all ensemble members in time (as is done in data assimilation cycles using the numerical model). In this case we also train a different model for each ensemble member which is also indicated in the equation.

$$\mathbf{x}^f_{(j),k} = \underbrace{\alpha_{(j)}\left(\mathbf{x}^a_{(j),k-1}\right)\mathbf{x}^a_{(j),k-1} + \beta_{(j)}\left(\mathbf{x}^a_{(j),k-1}\right)}_{\widehat{\mathcal{M}}_{(j)}\left(\mathbf{x}^a_{(j),k-1}\right)} + \boldsymbol{\eta}_{(j),k},$$

14. Line 182. "have" should be "has".

Thank you very much. We corrected this in the revised version of the manuscript.

15. Line 183. "Secondly, model errors associated with the local linear regression are considered and contribute to the forecast ensemble spread." Very unclear, please explain better.

We proposed replacing the following sentence: "Secondly, model errors associated with the local linear regression are considered and contribute to the forecast ensemble spread." by "Secondly, uncertainty in the local linear regression is considered as a robust estimation of model errors and contributes to increase the forecast ensemble spread (Lguensat et al. 2017 and Platzer et al. 2021)." to increase the clarity.

16. Line 204–205. What is the order of the Runge-Kutta? It is written "4-5". Then, is the model time step, dt = 0.1, equals to the assimilation interval?

Thank you very much. It is of order 4 and we corrected the text accordingly.

17. Line 215–216. "Figure 3 shows the mean CME for different values of $\lambda$". I think this sentence can be moved before when you started describing the figure.

We agree with the suggestion. We put the sentence at the beginning of the description of Figure 3 and modify it as follows: "Figure 3 shows the mean CME for different values of $\lambda$, for different configurations and, for the EnKF and the AnEnKF"

18. Line 221. x and x a should be bold.

We corrected this in the revised version of the manuscript.

19. Caption of Table 1. It should be "parentheses".

We corrected this in the revised version of the manuscript.

20. Line 267. Do you mean "latter"?

We corrected this in the revised version of the manuscript.

21. Lines 275–276. "Note that data assimilation cycles performed for each grid point are completely independent of their neighbours". This is very unclear. Please explain better, in particular about the concept of gridpoints independence that seems contradicting the multivariate nature of DA.

We agree with the reviewer. This Section was not clear enough and many key aspects of the implementation were not well described. Based on this comment and on the following comments we rewrite Section 3.2.1 Data driven model selection with SPEEDY. We hope that the new version of the section helps to clarify the implementation of AnDA with SPEEDY.

122. Lines 277–282. This part is also very unclear. You need to work it out better as it contains key details of your method. For instance, you need to construct your catalogue by

running the full model anyhow, isn't? Then I understand you search for analog within the 3x3 domain.

We agree with the reviewer. There is a significant computational cost associated with running the numerical model to generate the catalogs. We include the following sentence to stress this important point.

"It is important to note that the generation of the catalog brings a significant computational cost in this approach since it requires running the global numerical model once over a long period of time. However, we argue that for the implementation of this technique in real data applications, available long model simulations like those produced by the Coupled Model Intercomparison Project (Eyring et al. 2016) can be used. Moreover, the length of these catalogs are of the same order of magnitudes as the ones used in the idealized experiments with the SPEEDY model."

23. Line 285. Please explain from where the number of 250 analogs comes from the other experiments details.

We agree. This and other experimental settings have been chosen based on preliminary experiments. We add the following sentence in the revised version of the manuscript:

"These configuration settings have been chosen based on preliminary experiments performed over a limited number of local domains in which the sensitivity of the results to these parameters has been explored."

24. Line 286. What do you mean by "staggered"? What is the difference then between assimilation and observation frequency?

We agree that the expression is not clear. We modified that part of the text as follows:
"To increase the evidence associated with the local dynamics of the models the assimilation frequency is set to 24 hours. To take advantage of 6-hourly data, at each local domain, we perform four DA experiments which are run independently from each other starting at 00, 06, 12 and 18 UTC on the first day. These four DA cycles are performed over the same 3-years period."

25. Line 287. Please explain how do you get 4, 380 analysis cycles.

We agree this was not clear and the reason is partially related to the lack of clarity in the description of the experiment (as noted in the previous comment). We modify the text as follows to improve the clarity:

"The analysis obtained from these experiments are merged to obtain a total of 4,380 analysis cycles over the three-years assimilation period (4 DA experiments x 1095 cycles each)"

26. Line 312. I suggest that you include the equation as a formula in a generic way valid for RMSE and CME.

We agree with the reviewer, we add equations 17 and 18 to introduce the percentage difference in the RMSE and the CME.

27. Lines 324–326. I overall agree of what is said right before and it seems to me that these conclusions are in line with those Metref et al, 2019 and Carrassi et al, 2017. However, I do not understand the lines here: what large errors are you talking about?

We agree with the reviewer, the sentence was neither clear, nor correct. We propose to rephrase that part of the text as follows:
"A mismatch between the forecast and the observations will contribute less to the CME when the forecast uncertainty is correctly specified. We achieved this by the implementation of AnDA with adaptive multiplicative inflation. This takes into account the effect of stochastic errors in the initial conditions and its amplification due to the chaotic nature of the system which is not explicitly considered in the RMSE metric."

28. Line 397. Do you mean "weighting"?

Yes, we corrected this in the revised version of the manuscript.

29. Conclusion. I think the Authors results is also potentially very relevant for the so called "super-modelling" approach, whereby a dynamic combination of imperfect models is built. There the issue on how to "weight" the different models entering the combination is very important (see e.g. Schevenhoven et al. (2019)) and their approach may provide a ranking of model quality.

Thank you for this suggestion. This is certainly an interesting future direction for our research. We add the following comment in the discussion section:

"Another possible application of the AnDA-CME framework is in the context of weighted supermodels (Schevenhoven et al. 2019, Schevenhoven and Carrassi 2022) which provides a way to combine the time derivatives of different models resulting in improved short range and long range predictions."

References
Schevenhoven, F., F. Selten, A. Carrassi, and N. Keenlyside, Improving weather and climate predictions by training of supermodels, Earth System Dynamics, 10 , 789–807, 2019.

---

## Author Comment (AC2)

**Reviewer 2**

The submitted paper discusses an interesting, novel and computationally efficient approach to model evaluation which builds on concepts of ensemble Kalman filtering and analog approaches to build skill scores which indicate some degree of skill in identifying model errors when assessed in a perfect model framework. The technique is demonstrated for a highly idealised case (the Lorenz 63 model) and an intermediate complexity climate model.

The paper is well written, and novel. My opinion is that it should be published with only minor edits.

We would like to thank the reviewer for the comments that helped to improve the clarity of the manuscript.

Minor comments:

1 The approach described here is acceptable as a proof of concept - however it is likely not an optimal use of the data used in the training simulation used to assemble the analogs. In particular, the use of only small-scale information in the construction of analogs is discarding valuable information which would be represented in the covariance structure of the model output. The need to minimise the state space of model in order to find acceptable analogs is clear - but my suspicion is that a compression of state space which preserves elements of large scale covariance (such as PCA), rather than isolated regional analyses, would be even more effective.

We would like to thank the reviewer for bringing this interesting point. We agree with this idea and we add the following sentence in the discussion section:

"Implementing the combination of CME and AnDA in real-data cases brings additional challenges. For instance, in this work the application of the analog regression technique to a high-dimensional problem is achieved by using local domains. However, this approach does not take advantage of the covariance structure of the model output. This structure could be retrieved through a principal component analysis which may allow the implementation of the analog regression in a low dimensional space while keeping the main aspects of large scale circulation patterns."

2. A discussion of the dependency of performance on training run length for the analog would be useful, compared to a forecast-based approach (in the Lorenz case), and in terms of the ability to distinguish model errors (for SPEEDY).

A sensitivity experiment to the length of the catalog has been performed with the Lorenz 63 system, with or without data assimilation (see Table 1 for details). We also add a discussion about the selection of the length of the catalog in the SPEEDY experiments:

"AnDA experiments are conducted assimilating the observations generated from the last three years of the TRUE simulation. The catalogs for the analog forecasting are constructed

from the first 25 years of the RH08, RH07, and TRUE model runs and 250~analogs are used for the forecast.

In the SPEEDY experiments, the catalog contains over 36.000 samples (which is almost 4 times the size of the largest catalog which we tried with the Lorenz model). Although the local state space dimension that we used in SPEEDY is much larger (27 grid points), we argue that since there are substantial correlations among the state variables, the effective dimension can be significantly smaller.

The number of ensemble members is 30. To increase the evidence associated with the local dynamics of the models the assimilation frequency is set to 24 hours. To take advantage of 6-hourly data, at each local domain, we perform four DA experiments which are run independently from each other starting at 00, 06, 12 and 18 UTC on the first day. These four DA cycles are performed over the same 3-years period. These configuration settings have been chosen based on preliminary experiments performed over a limited number of local domains in which the sensitivity of the results to these parameters has been explored.

The analysis obtained from these experiments are merged to obtain a total of 4,380 analysis cycles over the three-years assimilation period (4 DA experiments x 1095 cycles each). It is important to note that the generation of the catalog brings a significant computational cost in this approach since it requires running the global numerical model over a long period of time. However, we argue that for the implementation of this technique in real data applications, available long model simulations like those produced by the Coupled Model Intercomparison Project (Eyring et al. 2016) can be used. Moreover, the length of these catalogs are of the same order of magnitudes as the ones used in the idealized experiments with the SPEEDY model."

4. Though the authors have demonstrated that CME provides a generally improved regional assessment of model error, this is not universally the case - especially for RH08, where ME provides a stronger signal in a number of regions. A short discussion on regions where this occurs, and potentially why, would be useful.

We agree with the reviewer, Figure 8 shows some areas where in fact RMSE performs better than CME at selecting the perfect model. We add a comment on this on the result section as well as a brief discussion of the possible cause of this issue.

"Although CME usually performs better than the RMSE at identifying the correct model, this is not always the case (see for example in Figure 8 how the probability of correct identification is larger for the RMSE than for CME near the Equator). This result may be due to an overestimation of the forecast error covariance $\Sigma^f$, computed within the analog procedure. Indeed, as explained in Eq. (11), an augmentation of this error matrix implies a diminution of the CME, and thus a decrease of performance of this metric."

---

## Author Response (AR2)

Dear Authors,

Thank you very much for the revised version of the manuscript and for addressing my original concerns very satisfactorily.
I think the manuscript is now ready to accepted and I do only have a few very minor remarks:

We would like to thank Reviewer 1 for the very constructive and insightful comments that helped to improve this manuscript a lot.

- in the equality Q=0 I would suggest to use boldface on both sides of the equality

Thank you very much. We corrected this in Line 211 of the new version of the manuscript.

- in response to my comment #11 the first sentence of the new text does not seem to be correct. What do you mean by "... model meaning." ?

This was our mistake. The word "meaning" should have been removed. We removed that word from the revised version of the manuscript in Line 149.

- in relation to my original comment #16: what is the time step and the observation frequency?

Sorry for not fully answering this point. We modify the sentence starting on Line 207 as follows: "In practice, the Lorenz model is integrated using a Runge-Kutta 4 scheme with a time step $dt$=0.1, and the three components are observed and assimilated at every model time step."

Besides the above, I am very much in favour of accepting the manuscript for publication on GMD.